# Distinct Ground Reaction Forces in Gait between the Paretic and Non-Paretic Leg of Stroke Patients: A Paradigm for Innovative Physiotherapy Intervention

**DOI:** 10.3390/healthcare9111542

**Published:** 2021-11-12

**Authors:** Zoe Mass Kokolevich, Erik Biros, Oren Tirosh, Jacqueline Elise Reznik

**Affiliations:** 1“Eshel Avraham” Centre for Special Needs Adults, Ezer Mizion, Bnei Brak 51553, Israel; zoe.masskokolevich@my.jcu.edu.au; 2College of Medicine and Dentistry, James Cook University, Townsville, QLD 4811, Australia; erik.biros@jcu.edu.au; 3Biomechanics School of Health Sciences, Swinburne University of Technology, Melbourne, VIC 3122, Australia; otirosh@swin.edu.au; 4Australian Institute of Tropical Health and Medicine, James Cook University, Townsville, QLD 4811, Australia

**Keywords:** stroke, hemiplegic gait, asymmetry, physiotherapy

## Abstract

This case report study aims to identify the differences in the ground reaction forces (GRF) placed on the forefoot, hindfoot, and entire foot between the paretic and non-paretic legs in two stroke patients to identify potential targets for improved physiotherapy treatment. A digital gait analysis foot pressure insole was fitted inside the participants’ shoes to measure the percentage of body weight taken during the stance phase, and the vertical GRF of the two subjects are reported in this paper. Both patients presented noteworthy differences in gait parameters individually and between their paretic and non-paretic legs. The trend shows a decreased percentage of body weight on the paretic forefoot and hindfoot, although the percentage bodyweight placed on the entire foot remained similar in both feet. The gait patterns shown were highly individual and indicated that both legs were affected to some degree. These findings identify key motion targets for an improved physiotherapy treatment following a stroke, suggesting that physiotherapy treatment should be targeted and individually tailored and should include both extremities.

## 1. Introduction

The characteristics of normal human walking include a cyclical upward vertical shift of the body’s centre of gravity that moves through the cycle of vertical motion with each step or two cycles during each stride (100% gait cycle) [1]. In addition, the vertical force from the ground during the stance phase (when the leg is on the ground) has a characteristic double hump (Figure 1A). The first hump relates to weight acceptance onto the foot when the body’s downward velocity is arrested (absorption phase). The second hump is due to push-off from the toes to propel the body forward (propulsion phase) [2].

An abnormal gait pattern is one of the most significant contributions to functional disability post-stroke [3,4]. Ground reaction force (GRF) represents the force exerted by the ground on a body in contact with it [5] and is commonly used to assess lower limb support function in stroke patients. In this context, spastic hemiparesis represents the primary motor impairments post-stroke where muscles may be both weak and spastic. Spastic hemiparesis may result from lesions in multiple areas of the brain, including those controlling movements of the upper limb, trunk, and lower limb, with a resultant broad spectrum of gait abnormalities [6]. In addition, the published literature discusses that improved weight transfer through the paretic leg following a stroke may improve gait symmetry, leading to a more functional gait [7,8,9].

This study reports two distinct bodyweight distribution gait patterns following stroke, for both paretic and non-paretic legs, and discusses a need for a tailored physiotherapy intervention to correct the potentially debilitating complications of an abnormal gait. The gait patterns in this report were characterised by the vertical GRF measured at the hindfoot, forefoot, and entire foot when walking at comfortable walking speeds. The vertical GRF was measured using plantar pressure insoles from the SmartStep™ Gait System Andante Medical Devices (Israel) and reported as a percentage body weight.

## 2. Presentation of Two Cases

We present the vertical GRF of two patients with ischemic stroke verified through the computerised tomography (CT) scans in which percentage body weight (% BW) placed on the paretic and non-paretic legs was measured using a digital gait analysis tool (Figure 1B,C). Neither patient had any history of neurological problems that could affect walking, such as Parkinson’s or Alzheimer’s disease. Figure 2 represents the output of the entire GRF from the SmartStep™ Gait System of the paretic leg of two stroke patients, a 59-year-old and a 73-year-old, A and B, respectively. Despite the apparent similarity in the patients’ neurological gait patterns and the similar time frame (6 months post-stroke), it is clear that the GRF patterns are dissimilar; thus, different intervention approaches should be considered.

The entire foot vertical GRF can be further analysed and segmented to GRF at the hind and forefoot. Figure 3A shows the average of hind-, fore-, and entire foot weight distributions from eight gait cycles of a 59-year-old woman, six months after her first episode of stroke. She presented with a relatively high %BW on her paretic hindfoot (70.8%) but only 95.2% on her forefoot during push-off. This gait pattern may indicate that she has difficulty differentiating between the heel strike and toe-off.

The gait cycle of the second study participant, a 73-year-old woman six months post-stroke, is shown in Figure 3B. She presented with a poor heel strike and, in turn, a poor push-off. The % BW placed on the hindfoot and forefoot of her paretic leg during the gait cycle was 33.7% and 76.2%, respectively. In addition, her non-paretic leg also presented with reduced % BW on the hindfoot (47.2%) but 105.9% on her forefoot during push-off. This pattern indicates that a poor heel strike affects the acceleration forward onto the forefoot and affects the toe-off. In addition, this deviation can affect the foot trajectory and placements of the opposite foot and heel strike, affecting the % BW on the opposite forefoot, thus affecting acceleration forwards.

## 3. Discussion

The individual analyses of the gait patterns show how altering the loading weight on the leg may alter the motor output pattern. In addition, we noted that a gait abnormality in the paretic leg could affect the foot trajectory and placements of the opposite non-paretic foot, thereby affecting the entire gait cycle. Thus, our findings suggest that individual gait parameters should be measured in all patients following stroke, and physiotherapy treatment should focus on appropriate foot contact for paretic and non-paretic lower limbs [7]. This approach represents an addition to older techniques such as those advocated by Wall and Turnbull, [10], where these authors discussed that a symmetrical gait might be achieved by focusing on the paretic leg only during the stance phase, assuming that the non-paretic leg may likely have strategies to compensate for the altered foot contact pattern on the paretic leg.

Our findings are, however, inherently associated with some limitations. In particular, each leg was assessed only once, without a trial run. It is acknowledged that performing more trials may have improved the results’ reliability [11]; however, the physical exhaustion of patients, affecting the results, could not be excluded. In addition, the digital gait analysis tool used was designed to measure each foot individually. In order to measure the steps of the gait cycle concurrently, two systems were required. This setup adds variation to the results, as the average measurements of the same gait cycles are not recorded. Finally, participants were asked to walk at their most comfortable speed, possibly adding inconsistencies to the results. The speed at which an individual walks can alter the clinical presentation of gait [12].

## 4. Conclusions/Clinical Message

This study provides supportive evidence that the entire gait pattern in the paretic and non-paretic legs may be affected following a stroke, indicating that a more patient-centered approach to physiotherapy intervention is needed.

## Figures and Tables

**Figure 1 healthcare-09-01542-f001:**
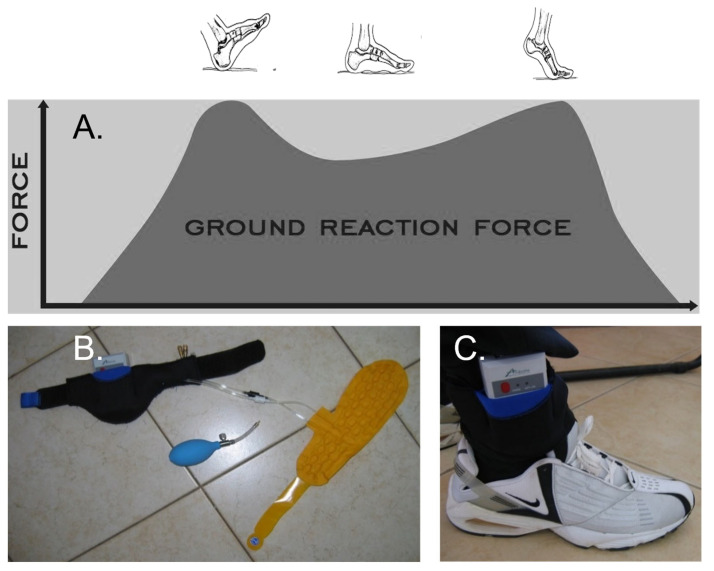
Ground reaction forces during normal gait (a butterfly diagram) and digital gait analysis tool. (**A**) The force is minimal at heel-strike, rises to a maximum soon after, falls a bit during the middle stance, rises again at toe-off, and finally falls to complete the gait cycle. (**B**) The SmartStep™ Gait System insole, Andante Medical Devices (Israel). The tool comprises the insole (yellow) with two sensors; the pump inflates the insole and the central processing unit to record gait parameters. (**C**) The insole was inserted into a shoe, and a participant wore the whole assembly.

**Figure 2 healthcare-09-01542-f002:**
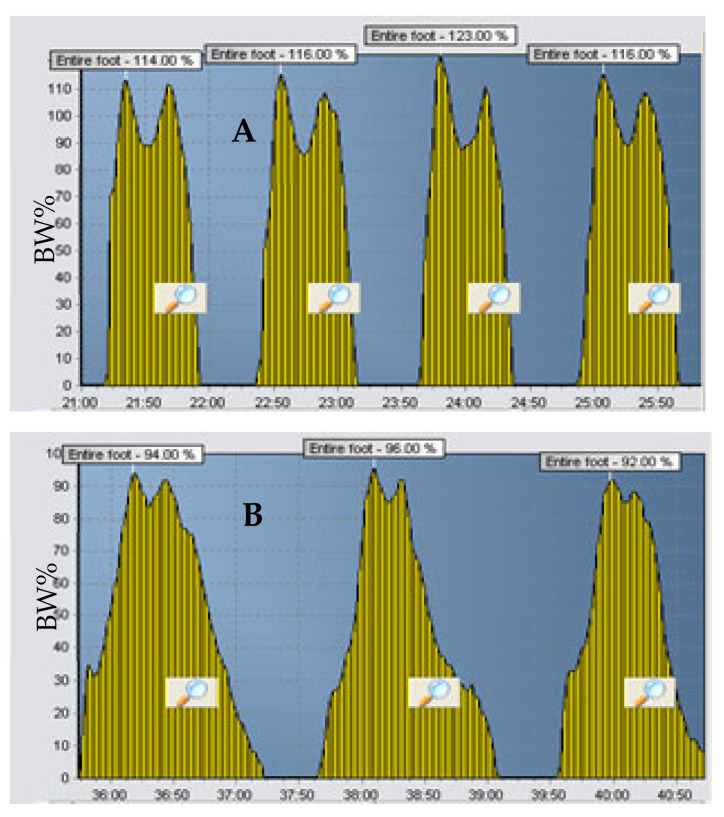
Unusual vertical GRF of two patients’ entire paretic gait cycle six months post-stroke (**A**,**B**). Vertical GRF is presented as a percentage of body weight (BW%).

**Figure 3 healthcare-09-01542-f003:**
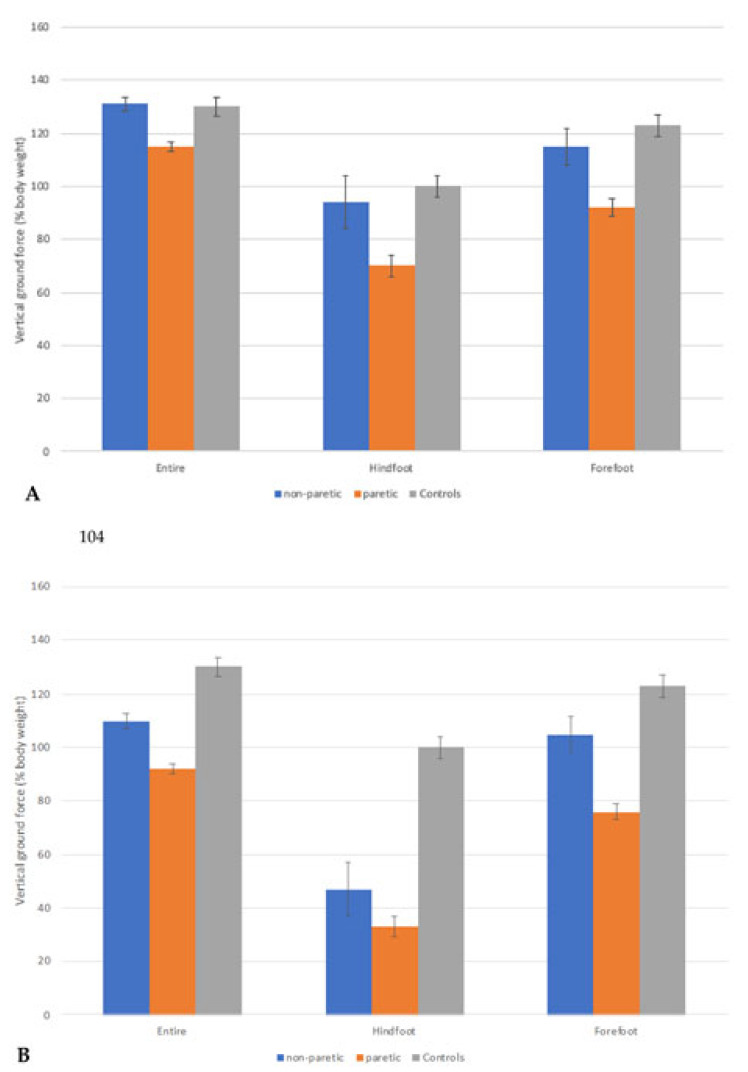
Means and standard deviations (error bars) of vertical GRF (% bodyweight) from eight gait cycles during comfortable walking for 59- and 73-year-old stroke patients (**A**,**B**), respectively. Outcomes are presented for the paretic and non-paretic legs—the vertical GRF of healthy gait is also included as a visual reference of “normality”. The GRF of healthy gait was derived from eight elderly volunteers recruited from the community who had no medical conditions that could have limited their walking ability.

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
