# Peer review of "Distinct Ground Reaction Forces in Gait between the Paretic and Non-Paretic Leg of Stroke Patients: A Paradigm for Innovative Physiotherapy Intervention"

_healthcare, 2021, doi:10.3390/healthcare9111542_

Round 1

Reviewer 1 Report

The article presents two case reports with gait disturbances after a stroke.

Both case reports show the wide range of gait disturbances after a stroke.

The authors suggest performing a gait analysesi for each patient after a stroke.

However, the consequences resulting from such an analysis referring to treatment should be presented in more detail. 

Limitations are discussed sufficiently.

Author Response

Response to Reviewer 1 Comments

Point 1: The article presents two case reports with gait disturbances after a stroke.

Point 2: Both case reports show the wide range of gait disturbances after a stroke.

Point 3: The authors suggest performing a gait analysis for each patient after a stroke.

However, the consequences resulting from such an analysis referring to treatment should be presented in more detail. 

Response to Point 3: As requested changes have been made to present the data and results in a clearer way. Additional figures have been added.

Point 4: Limitations are discussed sufficiently.

Reviewer 2 Report

This is a well written case report that addresses an important topic like the biomechanical assessment of both limbs (paretic and non-paretic) after stroke to drive customized physiotherapy treatment. However, I have several concerns, especially on the main conclusions and results presented. For instance, figure 1A represents typical vertical GRF, but this was not assessed with the same digital gait analysis tool at least in one healthy volunteer, which could be used to validate the tool. Moreover, figure 2 just represents GRF of the entire foot (from the paretic leg). The reader cannot see individual components from hindfoot or forffot, neither from the non-paretic leg. Still, in my opinion, some flaws can be corrected.

General comments

Although there is few space for references in case reports, at the end of my review I wrote 3 references that might be of help to improve the manuscript and specially to understand some of my comments.

This paper focus on GRF, which is one of the main kinetics variables assessed in gait biomechanics. Therefore, the title seems misleading, as gait parameters can include kinematics and spatiotemporal parameters. I suggest the authors to be more specific in the title.

In the abstract, the mention to “three gait rockers” comes with no previous explanation. Please take into account that Healthcare has readers from very different areas, that might not be familiar with the term.

The authors might clarify that GRF is a 3D force, usually assessed in terms of three components: 1) the vertical one, which is the largest of the three components; 3) the anteroposterior (related with braking and propulsion); and 3) the mediolateral (see DeLisa, 1998). It seems that the authors assessed the vertical component, although this was not mentioned in the manuscript. Moreover, continuous line in Figure 1A represents the typical vertical GRF while the dashed line represents the anteroposterior GRF. The latest is never mentioned in the manuscript. If authors present friction force (or braking force), they could also add the caption of propulsion (positive part of the dashed line). In fact, paretic propulsion (percentage of total propulsion force performed by the paretic leg) has been used to assess stroke patients and predict walking recovery following customized rehabilitation (Bowden et al, 2010; Barroso et al, 2017).

Lines 19-21. This is related with the previous comments. In opinion, this conclusion is related with the fact that patients might present reduced paretic propulsion. It would be interesting to check this component of the GRF, as it explicitly compares both sides (paretic and non-paretic).

Lines 51-53 – It might be difficult to understand that these are two rare and distinct gait patterns following stroke if the authors do not present “normality” after stroke. In my opinion, each patient has different patterns and should be treated individually. In other words, the authors should provide a better case to justify that these are rare patterns.

Caption from Figure 2. Where can the reader see information from the non-paretic side? And what about the individual components for hindfoot and forefoot? Can these be presented as supporting material?

Minor comments:

Figure 2 need some editing.

Lines 42-43 – The beginning of the quotation marks is missing.

References:

DeLisa, J. A. (Ed.). (1998). Gait analysis in the science of rehabilitation (Vol. 2). Diane Publishing.

Bowden, M. G., Clark, D. J., & Kautz, S. A. (2010). Evaluation of abnormal synergy patterns poststroke: relationship of the Fugl-Meyer Assessment to hemiparetic locomotion. Neurorehabilitation and neural repair, 24(4), 328-337.

Barroso, F. O., Torricelli, D., Molina-Rueda, F., Alguacil-Diego, I. M., Cano-de-la-Cuerda, R., Santos, C., ... & Pons, J. L. (2017). Combining muscle synergies and biomechanical analysis to assess gait in stroke patients. Journal of Biomechanics, 63, 98-103.

Author Response

Response to Reviewer 2 Comments

Point 1: References that might be of help to improve the manuscript and specially to understand some of my comments.

Response to Point 1: We thank the reviewer for these comments. We have read the references suggested and have added the most recent one to our paper.

Point 2: This paper focus on GRF, which is one of the main kinetics variables assessed in gait biomechanics. Therefore, the title seems misleading, as gait parameters can include kinematics and spatiotemporal parameters. I suggest the authors to be more specific in the title.

Response to Point 2: As suggested the title was revised to “Distinct ground reaction forces in gait between the paretic and non-paretic leg of stroke patients: a paradigm for innovative physiotherapy intervention”

Point 3: In the abstract, the mention to “three gait rockers” comes with no previous explanation. Please take into account that Healthcare has readers from very different areas that might not be familiar with the term.

Response to Point 3: Abstract was revised as suggested “… A digital gait analysis foot pressure insole was fitted inside the participants' shoes to measure the percentage of body weight taken through the body support stance phase, and the vertical ground reaction forces of the two subjects are reported in this paper……”.

Point 4: The authors might clarify that GRF is a 3D force, usually assessed in terms of three components: 1) the vertical one, which is the largest of the three components; 3) the anteroposterior (related with braking and propulsion); and 3) the mediolateral (see DeLisa, 1998). It seems that the authors assessed the vertical component, although this was not mentioned in the manuscript.

Response to Point 4: As suggested manuscript was revised to explain that only vertical ground forces were measured. See last paragraph of the introduction line 63 “The gait patterns in this report will be characterised by the vertical GRF measured at the hindfoot, forefoot, and entire foot when walking at comfortable walking speed. The vertical GRF were measured using plantar pressure insoles from the SmartStep™ Gait System Andante Medical Devices (Israel).”

Point 5: Moreover, continuous line in Figure 1A represents the typical vertical GRF while the dashed line represents the anteroposterior GRF. The latest is never mentioned in the manuscript. If authors present friction force (or braking force), they could also add the caption of propulsion (positive part of the dashed line). In fact, paretic propulsion (percentage of total propulsion force performed by the paretic leg) has been used to assess stroke patients and predict walking recovery following customized rehabilitation (Bowden et al, 2010; Barroso et al, 2017).

Response to Point 5: The figure was revised to include only the vertical ground reaction forces.

Point 6: Lines 19-21. This is related with the previous comments. In opinion, this conclusion is related with the fact that patients might present reduced paretic propulsion. It would be interesting to check this component of the GRF, as it explicitly compares both sides (paretic and non-paretic).

Response to Point 6: Unfortunately, the pressure insoles measure only the vertical forces. We agree with the reviewer that anterior/posterior and medio/lateral forces may provide further insight. We plan to measure these forces in the future using force-plates in our gait laboratory.

Point 7: Lines 51-53 – It might be difficult to understand that these are two rare and distinct gait patterns following stroke if the authors do not present “normality” after stroke. In my opinion, each patient has different patterns and should be treated individually. In other words, the authors should provide a better case to justify that these are rare patterns.

Response to Point 7: We included another figure (Figure 3) that shows the weight distribution of the hind-, fore-, and entire foot of both paretic and non-paretic legs. The figure also includes average weight distribution of healthy controls.

Point 8: Caption from Figure 2. Where can the reader see information from the non-paretic side? And what about the individual components for hindfoot and forefoot? Can these be presented as supporting material?

Response to Point 8: As suggested captions were revised. In addition, Figure 3 was added to include hindfoot and forefoot data.

Minor comments:

Point 9: Figure 2 need some editing.

Response to Point 9: Both figure and captions in Figure 2 were revised

Point 10: Lines 42-43 – The beginning of the quotation marks is missing.

Response to Point 10: Quotation mark was deleted.

Point 11/12: References:

DeLisa, J. A. (Ed.). (1998). Gait analysis in the science of rehabilitation (Vol. 2). Diane Publishing. 

Bowden, M. G., Clark, D. J., & Kautz, S. A. (2010). Evaluation of abnormal synergy patterns poststroke: relationship of the Fugl-Meyer Assessment to hemiparetic locomotion. Neurorehabilitation and neural repair, 24(4), 328-337.

Response to Point 11: We did not use these two references as that are more recent have been used.

Barroso, F. O., Torricelli, D., Molina-Rueda, F., Alguacil-Diego, I. M., Cano-de-la-Cuerda, R., Santos, C., ... & Pons, J. L. (2017). Combining muscle synergies and biomechanical analysis to assess gait in stroke patients. Journal of Biomechanics, 63, 98-103.

Response to Point 12: This reference has been added on line 53.

Round 2

Reviewer 2 Report

This new revision led to an improved manuscript. I want to thank the authors for addressing my major concerns (although I was not able to see the other reviewer(s) comments).

Minor comments that do not need further review:

Lines 17-18: It seems that there is a word missing between “body support” and “stance phase”. Maybe “during”?

Figure 1. In this case, the title for xx axis should be ‘Stance phase (%)’ instead of ‘Gait Cycle (%)’. During swing phase (the remaining part of the gait cycle), there is no GRF (leg is not anymore on the ground). In this sense, the caption of Figure 1A should be edited. Specifically, there is no left (red) or right (blue) in the figure and the caption should refer to the stance phase instead of the gait cycle.

Figure 1C. Caption is missing.

Figure 2. Please clarify what ‘N/BW%’ is. Maybe normalized body weight? I would suggest the authors to use the same nomenclature across the manuscript (e.g., using N/BW% or other acronym in Figures 2 and 3).

Caption of Figure 3, line 112. Means and standard deviation (error bars) of vertical ground reaction force (% body weight). “of” is missing.

Figure 3. Where does the information from healthy subjects come from? This should be explicitly mentioned in the text.

The acronym GRF should be used across the text to avoid the repetition of “ground reaction force”.

Please clarify why these 2 patients are considered similar. Is there because they suffered a stroke 6 months before the study? Or because the vertical GRF of the entire foot is considered by the authors as similar (although there is a difference of around 20%)?
